# Enthalpy-Entropy Compensation Effect in Saturated Solutions on an Example of Polynuclear Aromatics According to Thermodynamics at Melting Temperature

**DOI:** 10.3390/e25010055

**Published:** 2022-12-28

**Authors:** Andrzej Mianowski, Grzegorz Łabojko

**Affiliations:** Institute for Energy and Fuel Processing Technology, Zamkowa Street 1, 41-803 Zabrze, Poland

**Keywords:** solubility, aromatic hydrocarbons, enthalpy, entropy, activity, EEC

## Abstract

A thermodynamic the influence of temperature on the logarithm of the considered quantity is expressed by bifunctional functional terms (1/*T*, ln*T*). For this purpose, the Apelblat & Manzurola (A&M) equation was used for extended model dissolution analysis of 12 aromatic hydrocarbons in tetralin and decalin vs. temperature for saturated solutions. The A&M equation was found to be thermodynamically compensatory in the sense of Enthalpy-Entropy-Compensation (EEC) while limiting melting temperature $\;T_{m}=\frac{{∆}_{m}H}{{∆}_{m}S}.$ The coefficients for the functional terms $A_{1}\;$ vs. $A_{2}$ are a linear relationship, with a slope called the compensation temperature $T_{c}$, as ratio of average enthalpy to average entropy. From this dependence, it has been shown that the approximation of $∆c_{p}={∆}_{m}\bar{S}$ is justified, also assuming the average entropy. Regarding the term representing the activity coefficients, modifications to the A&M equation were proposed by replacing the intercept and it was shown that the new form correctly determines ${∆}_{m}H.$ However, the condition is that the molar fraction of the solute exceeds *x* > 0.5 moles. It has been shown that the simplest equation referred to van ’t Hoff’s isobar also allows the simultaneous determination of enthalpy and entropy, but these quantities do not always come down to melting temperature.

## 1. Introduction

In some areas of chemistry and chemical engineering, temperature-dependent quantities are expressed as the logarithm of that quantity to the reciprocal of absolute temperature. In general, what we mean is a chemical or phase equilibrium constant. A relationship containing temperature expressed in two functional terms, 1/*T* and ln (*T*), is often proposed. A classic example is the Rankine-Kirchhoff-Dupré equation [1] describing saturation vapor pressure vs. *T*, to which Romps attributes very high accuracy and usefulness [2].

The Arrhenius-Gutzmann equation is often modified in this way for changes in viscosity with temperature [3,4].

Depending on the nature of the phenomenon, the variable (1/*RT*) is associated with the activation energy or enthalpy, while the variable (±*R*ln [*T*]) can normalize the entropy.

Research of the dissolution process is of particular importance. This is due to the fact that, on one hand, from the galaxy of equations, we can distinguish those in which enthalpy dominates, only entropy, or both thermodynamic functions play comparable roles. On the other hand, the thermodynamic category is activity, although for experimental purposes it would be more preferable to use a solute mol fraction. The dissolution processes, due to the wide modeling possibilities, are of particular interest for the substantive assessment of the role of the bifunctional influence of temperature, especially since we can expect thermodynamic compensation, known as Enthalpy-Entropy-Compensation effect (EEC).

From the brief analysis of the literature on the problem of solid phase dissolution in various solvents, both weakly and moderately dissolving, one can see the gradual development of theories ranging from simple to ideal, binary to real, multi-component solutions with complex interactions at the molecular level. For example, items can be listed [5,6,7,8,9].

According to IUPAC [9] solubility is the analytical composition of a mixture or solution, which is saturated with one of the components of the mixture or solution, expressed in terms of the proportion of the designated component in the designated mixture or solution.

Generally, solubility models are derived from an equilibrium state in terms of [9]:(1)$$\frac{∆G^{\emptyset }\left(T,P\right)}{RT}=-ln\frac{a^{\left(\beta \right)}}{a^{\left(\alpha \right)}},$$
where the activity of the dissolved component is considered in its own phase (α) and dissolved (*β*).

The standard Gibbs free energy statement on the right side of Equation (1) can be used in many forms, also according to the Clarke-Glew concept [10] used in subsequent works [11,12,13,14].

Equation (1) for constant pressure and saturated solutions can be written in detailed form according to Clausius-Clapeyron:(2)$$\ln\left(a\right)=\int_{T}^{T_{m}}\;\frac{{∆}_{m}H}{R}d\left(\frac{1}{T}\right),\;P=\mathrm{const}$$

The activity of component (*a*) in a solution is strongly dependent on its activity coefficients, whose numerical values depend on the adopted model [15].

In historical terms, the first approach to the problem is due to the equations: Margules (1895), van Laar (1910) [16] and from 1964–Wilson [14,17,18,19,20,21,22], NRTL (Non-Random, Two-Liquid) [20,21,22], UNIQUAC (Universal Quasi-Chemical) [15,19,20,21,22,23,24] and UNIFAC (**UNI** QUAC **F**unctional-group **A**ctivity **C**oefficients, 1975) [14] based on the UNIQUAC algorithm or the latest ASOG (Analytical Solution of Groups, 1989) [25] using the Wilson equation. We also meet even newer, less known, e.g., MOQUAC [26] or older VLE [21,25].

## 2. Equations Describing Saturated Solutions in Terms of Temperature

In the creation of solutions in the liquid-solid phase system, a more convenient starting point for further considerations is a record in which thermodynamic functions for the fusion of the melting point temperature are exposed, as indicated in Equation (1):(3)ΔG(T)=−RTln(a)=ΔH−TΔS,

Because for $T=T_{m}$
(4)$$\Delta G\left(T_{m}\right)=0,\;\;\ln\left(a\right)=0,\;\;$$

then:(5)$${∆}_{m}S=\frac{{∆}_{m}H}{T_{m}},\;\;$$

If the thermodynamic functions are constant in the temperature range *T* →$\;T_{m}$, then Equation (3) can be written as:(6)$$\ln\left(a\right)=-\frac{{∆}_{m}H}{RT}+\frac{{∆}_{m}S}{R},$$

It is known that for an individual component in solution:(7)a=x∗γ,
where *x* is the molar fraction in solution saturated with a solid ($x=x_{2}^{sat})$ and the activity coefficient* γ * (*$\gamma =\gamma_{2}$*) also applies to the solid phase (which is often indexed in literature by the number 2) then Equation (6) can be represented in the form:(8)$$\ln\left(x\right)=-\frac{{∆}_{m}H}{RT}+\frac{{∆}_{m}S}{R}-\ln\left(\gamma \right),$$

Inserting the Equation (5) into the form Equation (8) we get the Schröder equation, which for the expression: ln(x∗γ) is an analogue of van ’t Hoff isobar:(9)$$\ln\left(x\right)=-\frac{{∆}_{m}H}{R}\left(\frac{1}{T}-\frac{1}{T_{m}}\right)-\ln\left(\gamma \right),$$

In Equation (3) it can also be assumed that thermodynamic functions are dependent on temperature according to Kirchhoff dependence, for $∆c_{p}=\mathrm{const}$ and $T<T_{m}$
(10a)$${∆}_{m}H=∆H+∆c_{p}\int_{T}^{T_{m}}dT,\;\;$$
(10b)$${∆}_{m}S=∆S+∆c_{p}\int_{T}^{T_{m}}dT/T$$

After integration and substitution into Equation (3) using Equation (5) and after ordering using Equation (7), we obtain the Hildebrand equation:(11)$$\ln\left(x\right)=-\frac{{∆}_{m}H}{R}\left(\frac{1}{T}-\frac{1}{T_{m}}\right)+\frac{∆c_{p}}{R}\left[\frac{T_{m\;}}{T}-1-\ln\left(\frac{T_{m}}{T}\right)\right]-\ln\left(\gamma \right),$$

In Equation (11) assuming that for ∆$c_{p}$ = 0, we get Equation (9) again.

For $∆c_{p}\approx \;{∆}_{m}S$ and using Equation (5) we get Hildebrand-Scott equation:(12)$$ln\left(x\right)=-\frac{{∆}_{m}S\;}{R}ln\frac{T_{m}}{T}-\ln\left(\gamma \right),$$

Using Equation (5) in Equation (9) we get we get an alternative expression to Equation (12) [27]:(13)$$ln\left(x\right)=-\frac{{∆}_{m}S\;}{R}\left(\frac{T_{m}}{T}-1\right)-\ln\left(\gamma \right),$$

According to [23] the approximate relationship $\left(∆c_{p}\approx \;{∆}_{m}S\right)\;\mathrm{too}$ often deviates from the identity.

In turn Equation (11) can be grouped differently:(14)$$\ln\left(x\right)=\frac{\Delta C_{p}T_{m}-{∆}_{m}H}{RT}+\frac{\Delta C_{p}}{R}lnT+\frac{{∆}_{m}H-\Delta C_{p}T_{m}\left(1+lnT_{m}\right)}{RT_{m}}-\ln\left(\gamma \right),$$

which according to [7] means at the same time with the proposal of Apelblat & Manzurola (A&M) [28,29,30,31,32,33].

Omitting the signs in Equation (14), the mathematical formula is obtained in the form of a three-parameter equation, when activity is determined by Equation (7):(15)$$\ln\left(a\right)=A_{o}+\frac{A_{1}}{T}+A_{2}\ln T,$$

One equation bypassing the need to determine activity coefficients is that in Buchowski et al., which by two parameters defines enthalpy of solution per mole of the solute [29,32,34].
(16)$$\ln\left(1+\lambda \frac{1-x}{x}\right)=\lambda \frac{∆H}{R}\left(\frac{1}{T}-\frac{1}{T_{m}}\right),$$

The Equation (16) was derived for compounds soluble through hydrogen bonding and the equation constants are determined by nonlinear regression [34].

The enthalpy of solution is therefore equal to the sum of the enthalpy of fusion and the enthalpy of mixing. However, since the enthalpy of mixing must equal zero for an ideal solution, it follows that the enthalpy of solution must equal the enthalpy of fusion of the solid at the given temperature [5].

## 3. Results

### 3.1. Objective of the Work

The development in the field of Clean Coal Technology, especially in the production of carbon liquids from a carbonized solid phase, is associated with the need to extend research on the solubility of aromatic compounds in selected non-electrolytes. The most important are the compounds found in coke tar or low-temperature tar, and thermodynamic analyzes of such systems in different solvents are necessary.

Summing up the possibilities of determining thermodynamic functions in the equilibrium dissolution process, it can be seen that:depending on the purpose of the study, ideal solutions are considered, where the activity coefficient is omitted (ln (*γ*) = 0) [5,27,35,36,37,38,39,40,41,42,43,44] in many works, mainly for pharmacy, the molar fraction *x* = $x_{2}^{\mathrm{sat}}$ on the left side of the Equation (11) is referred to as CLFR (Crystal-Liquid Fugacity Ratio) [39,41,43,44],systems in which the activity coefficients ln (*γ*) > 0 [5,6,7,8,9,15,16,17,18,19,20,21,22,23,24,25,26] are very extensive, usually point transition is omitted [45],for the A&M equation [28,29,30,31,32,33] it is worth considering how much it is possible to eliminate the troublesome determination of activity coefficients, associating this equation with Hildebrand’s theory.

The purpose of the work is to analyze the impact of knowledge of activity coefficients on the accuracy of determining basic thermodynamic functions at melting point temperature for binary regular solutions. The problem results from the fact that for the simplest experimental method, Equation (11) with possible additional physicochemical elements [46,47] is used for known values of basic thermodynamic functions (enthalpy, entropy, ∆c_P). Therefore, the question arises how knowledge of activity coefficients affects the estimation of selected thermodynamic functions.

From several publications [6,7,8,18,21,45,46] in the field of dissolution of aromatic compounds in similar solvents, the most physicochemical data were contained in [21]. The dissolution of 12 aromatic compounds (including those containing N, S, O heteroatoms) in tetralin and decalin (and their mixture) was studied. Complete studies were carried out for: biphenyl, fluorene, phenanthrene, acenaphthene, naphthalene, dibenzofuran, dibenzothiophene, thioxanthene, xanthene, carbazole, acridine and anthracene. 

The aim of this study is to use these experimental data to demonstrate the thermodynamic aspects of the A&M equation and to propose its modification in order to enrich it with activity coefficients. It is assumed that the measured values are sufficient to determine them. The Enthalpy-Entropy Effect (EEC) can play an important role in this.

### 3.2. The Thermodynamic Aspect of the A&M Equation

If the Equation (3) is presented in the form:(17)$$\frac{∆G}{T}=\frac{∆H}{T}-∆S,$$

then according to the Gibbs-Helmholtz equation
(18)$${\left[\frac{\partial \left(\frac{∆G}{T}\right)}{\partial \left(\frac{1}{T}\right)}\right]}_{P}=∆H,$$

what means for Equations (15) and (1) as $\frac{dln\left(a\right)}{d\frac{1}{T}}=-∆H/R$
(19)$$∆H=R\left(A_{2}T-A_{1}\right),$$

in particular for $\;T=T_{m}$
(20)$${∆}_{m}H=R\left(A_{2}T_{m}-A_{1}\right),$$

The second thermodynamic function is expressed directly by the formula from Equation (5), as: (21)$${∆}_{m}S=R\left(A_{2}-\frac{A_{1}}{T_{m}}\right),$$

By using Equation (3) again after differentiation we get: (22)$${\left(\frac{\partial ∆G}{\partial T}\right)}_{P}=-∆S,$$

what after writing: (23)$$-R\frac{d}{dT}\left[Tln\left(a\right)\right]=-∆S,$$

leads to the equation: (24)$$∆S=R\left[\ln\left(a\right)-\frac{A_{1}}{T}+A_{2}\right],$$

in which for *T* $=T_{m}$ and for condition Equation (4) formula Equation (21) is reproduced.

By implementing the structure of the A&M equation according to Equation (15) in Equations (18) and (23), expressions of thermodynamic functions depending on the basic measured relationships and its coefficients are created. According to Equations (19) and (20), the differences $\left|∆H-{∆}_{m}H\right|$ can be seen as the relation between the enthalpy at temperature T and the melting point temperature. In turn, entropy depends on the activity according to Equation (24). These properties are not shown by other equations containing only one functional element with respect to temperature.

The Gibbs free energy resulting from the Equations (19) and (24) is: (25)$$∆G=R\left(A_{2}T-A_{1}\right)-RT\left[\ln\left(a\right)-\frac{A_{1}}{T}+A_{2}\right]=-RTln\left(a\right),$$

and complies to Equation (3).

The components of Equation (25) may be used in the designation of the EEC:(26)$$R\left(A_{2}T-A_{1}\right)=RT_{m}\left[\ln\left(a\right)-\frac{A_{1}}{T}+A_{2}\right]+\beta_{o},$$
where for the system one compound in one solvent, the role of compensating temperature is taken by $T_{m}$, a $\beta_{o}$ = 0, when according to Equation (4), $T=T_{m}$, ln(a)=ln1=0.

After transforming, we get the equation:(27)$$\ln\left(a\right)=A_{1}\left(\frac{1}{T}-\frac{1}{T_{m}}\right)+A_{2}\left(\frac{T}{T_{m}}-1\right),$$

By inserting the coefficient $A_{2}$ in the expression $\left(\frac{T}{T_{m}}-1\right)≅ln\frac{T}{T_{m}}$, Equation (15) is reproduced. Thus, Equation (27) connects the elements of Equations (9), (12) and (13).

In this way, it was proved that Equation (15) is also EEC, because it contains in its meaning a straightforward linear relationship between enthalpy and entropy without intercept, when we operate with activity.

Thus, for Equation (15), enthalpies are expressed by Equation (20) and entropy by formula Equation (21). Ratios: Equation (20)/Equation (21) = $T_{m}$, and approximate also Equation (19)/Equation (24) = $T_{m}$ (Figure 1) which is consistent with the basic formula Equation (5). In the dissolution processes, an enthalpy invariant with respect to temperature is assumed, i.e., $\frac{d∆H}{dT}=0.$ Meanwhile, in Equation (19) an additional functional element appears, which means that $\frac{d∆H}{dT}=RA_{2}$. The comparison of Equation (14) and Equation (15) shows that $RA_{2}=∆c_{p},$ that is $\frac{d∆H}{dT}=∆c_{p}\;$ which is according to Kirchhoff’s law, Equation (10a). When $A_{2}=0$, then in Equations (15) and (27) the term in which entropy may appear disappears. There is a clear tendency to assign certain equations to experimental results. When studies are conducted near melting temperature, this is either $∆c_{p}\;≅0$ or/also $T_{m}/T\;≅1$ and the Hildebrand Equation (11) is reduced to the van ’t Hoff isobar, Equation (9).

Figure 1 shows an excerpt from further systematic analyzes in a typical EEC pattern. The enthalpy is calculated by the formula Equation (19) and the entropy by the formula Equation (24) for the coefficients: $A_{1}$ = 1060.6 K, $A_{2}$ = 9.764 (Equation (31)), in this case $T_{m}$ = 341.9 K and slope is 343.63 K.

The A&M equation represents the nature of the bifunctional relationship with temperature and therefore partial correlations imply EEC.

The nature of EEC can be explained by the system of correlation of coefficients at the functional members Equation (15) for more dissolved hydrocarbons in one solvent. It should be remembered at this point that the considerations relate to the activity and not the solute mol fraction.

### 3.3. Correlations Using Known Activity Coefficients

The correlation of activity coefficients according to the UNIQUAC model with model S-H for $l_{12}\;\neq 0$ indicates practical identity (see Figure 2).

Then, 6 types of equations were analyzed in which $x=x_{2}^{sat},\;\gamma =\gamma_{2}$:(28)$$\ln\left(x\right)=A_{o}+A_{1}\left(\frac{1}{T}-\frac{1}{T_{m}}\right)+A_{2}\ln\left(\frac{T}{T_{m}}\right),$$
(29)$$\ln\left(x\right)=A_{1}\left(\frac{1}{T}-\frac{1}{T_{m}}\right)+A_{2}\ln\left(\frac{T}{T_{m}}\right),\;\;A_{o}\;\equiv 0$$
(30)$$\ln\left(x*\gamma \right)=A_{o}+A_{1}\left(\frac{1}{T}-\frac{1}{T_{m}}\right)+A_{2}\ln\left(\frac{T}{T_{m}}\right),$$
(31)$$\ln\left(x*\;\gamma \right)=A_{1}\left(\frac{1}{T}-\frac{1}{T_{m}}\right)+A_{2}\ln\left(\frac{T}{T_{m}}\right),\;\;A_{o}\;\equiv 0$$
(32)$$\ln\left(x\right)=-\frac{{∆}_{m}H}{R}\left(\frac{1}{T}-\frac{1}{T_{m}}\right),\;\mathrm{intercept}\neq 0$$
(33)$$\ln\left(x*\gamma \right)=-\frac{{∆}_{m}H}{R}\left(\frac{1}{T}-\frac{1}{T_{m}}\right),\;\mathrm{intercept}=0,$$

The relationship in the form Equation (11) was also analyzed, which is not discussed in this paper due to the difficult-to-explain variable-sign discrepancies determined for $∆c_{P}$. 

Activity coefficients were used for the model extended S-H. For all cases Equations (28)–(31), determination coefficient for equations with intercept and without that expression is $R^{2},$ $\rho^{2}$ > 0.999 except for systems with carbazole in tetralin, where $R^{2},$ $\rho^{2}$ > 0.99, while for decalin systems only for dibenzofuran, carbazole, acridine $R^{2},$ $\rho^{2}$ ≥ 0.99 and in other cases $R^{2},$ $\rho^{2}$ > 0.999. Equation analysis also Equations (32) and (33) showed to their high significance ($R^{2}>0.99$) except for system carbazole + tetralin ($R^{2}$ = 0.97) according to Equation (32) and anthracene + tetralin ($\rho^{2}=0.92)$ according to Equation (33), which may be a surprise. For decalin solutions, deviations were observed for acridine in both Equations (32) and (33) ($R^{2}=\rho^{2}>$ 0.97).

From the point of view of the quality of equations, there is no doubt that these are approaches relevant for very high probability. The process enthalpy at temperature $T_{m}\;$ was determined for 6 of the equations analyzed (Table A1, Appendix A).

The determined coefficients of the Equations (28)–(31) were correlated in the form of a linear relationship: $A_{1}$ vs. $A_{2}$ assuming 4 × 12 = 48 pairs of coefficients for tetralin and identically for decalin. The graphical correlation image is shown in Figure 3. Figure 3c is the combined set of all data.

Thus, a comparison of the obtained calculation results presented in Table A1, using Equation (28) shows that for the criterion as the smallest difference in relation to ${∆}_{m}H$ known from the literature, they show equations in which the expression ln⁡(*x***γ*) and deviations from this remark (except for carbazole + decalin systems) do not exist. There are several systems that point to absurd values several times exceeding literature values when activity coefficients are not taken into account. This is especially visible for hydrocarbons containing heteroatoms dissolved in decalin.

In addition however, assuming the evaluation of the results presented in Table A1, according to other acceptance (or negation) criteria by compliance with melting temperature, Equations (28)–(31) for EEC determine the enthalpy and the associated entropy in $T_{m}$, taking into account the influence of the solvent.

Figure 3 indicates that for the studied population, a linear correlation analysis of the coefficients leads to the notation, where $T_{c}$ is compensation temperature,
(34)$$A_{1}=T_{c}A_{2}-\beta ,$$

From the comparison of the coefficients in Equation (14), the structures of the coefficients Equation (34) follow:(35a)$$A_{1}=T_{m}\frac{∆c_{p}}{R}-\frac{{∆}_{m}H}{R},$$
(35b)$$A_{2}=\frac{∆c_{p}}{R},$$

which implies for relations Equation (14) from Equation (20),
(36)$$T_{m}=\frac{A_{1}}{A_{2}}+\frac{{∆}_{m}H}{RA_{2}},$$

for each system one hydrocarbon–one solvent, $A_{1},\;A_{2}$ = *const*.

Thus Equation (36) is a transformation of Equation (20).

Summing up all Equation (36) for individual hydrocarbons, and the variables are $T_{m}$ then we get the Equation (37),
(37)$$T_{c}=\frac{A_{1}}{A_{2}}+\frac{{∆}_{m}\bar{H}}{RA_{2}}=const,$$

for many compounds dissolved in solvent, $A_{1}$ vs. $\;A_{2}.$

Equation (37) means, that in Equation (34) $\beta =\frac{{∆}_{m}\bar{H}}{R},$ and the compensation temperature replaces the arithmetic mean of all melting temperatures. In both Equations (36) and (37) for *A*_1_ = 0, the product of *RA*_2_ is related to the heat capacity, and in case of Equations (36) and (5) with entropy. Thus, Equation (36) is related to individual hydrocarbons (Figure 1):(38)$$T_{m}=\frac{{∆}_{m}H}{RA_{2}}=\frac{{∆}_{m}H}{{∆}_{m}S},\;\mathrm{for}\;A_{1}=0,$$

Equation (37) applies to many compounds dissolved in solvent, and from the view presented in Figure 3c it follows that the slope is:(39)$$T_{c}={∆}_{m}\bar{H}/RA_{2},\;\mathrm{for}\;A_{1}=0$$

and directly means:(40)$$T_{c}={∆}_{m}\bar{H}/{∆}_{m}\bar{S},$$

Equation (40) defines the thermodynamic aspect of the compensation temperature analogously to the approach acc. to Equation (5) for the considered population of 12 hydrocarbons in two solvents.

The EEC analysis for hydrocarbons without solvent is shown in Figure 4.

In particular, Figure 4b is a proof of the validity of Equation (38) with entered Equation (39) when $RA_{2}={∆}_{m}\bar{S}$. Figure 4a is only a graphical representation of the EEC relationship for pure hydrocarbons.

The analysis of Figure 3c shows that the slope as the coefficient $A_{2}\;$is variable-sign and is within the maximum range $A_{2}\in \;$[−94;124], so the product ($RA_{2})$ for these hydrocarbons is well over the range [21] $∆c_{p}=1.4-39.8\;$J(mol*K)^−1^, average 16.04 J(mol*K)^−1^.

According to [21] for the analyzed hydrocarbons ∆$c_{p}$ is below 40 J*(mol*K)^−1^, pure solvents have much higher heat capacity, tetralin $c_{p}\;$= 217.5 J*(mol*K)^−1^ (*T* = 298.15 K) [48] and decalin $c_{p}$ = 229.17 J*(mol*K)^−1^ (*T* = 298.15 K) [49]. 

The use of this analysis authorizes the formulation of a conclusion about equality such that $∆c_{p}$ is a substitute for entropy [50], but it is correct for averaged values from a given population.

In the considered system of the solubility of 12 hydrocarbons, it was established that with a good approximation, the apparent compensation temperature and the enthalpy are arithmetic means of melting temperatures and individual enthalpies. In particular, Equation (40) captures this effect. 

If average temperature melting is reduced to $T_{c}$ then we get the average entropy acc. Equation (40): ${∆}_{m}\bar{S}$ =$\;{∆}_{m}\bar{H}/T_{c}$ = 24,556.2/334.09 = 73.5 J*(mol*K)^−1^ (equation under Figure 3c, ${∆}_{m}\bar{H}=R*intercept)\;$and for data [21] we obtain ${∆}_{m}\bar{S}$ = 53.61 J*(mol*K)^−1^ or ${∆}_{m}\bar{S\;}=\frac{1}{12}\sum^{\;}\frac{{∆}_{m}H}{T_{m}}=$53.43 J*(mol*K)^−1^. The average value from two intakes: for hydrocarbons in coke oven tar (as it is in this paper) ${∆}_{m}S=$ 57 J*(mol*K)^−1^, and the resulting from Trouton rule ∆S= 88 J*(mol*K)^−1^ [39,44].

According to analyses [27,39,44] for entropy of 12 selected hydrocarbons tested in [17] ${∆}_{m}S=$ 44.25–64.87 J*(mol*K)^−1^, with the average value equal to 53.35 J*(mol*K)^−1^ (very high agreement with [39]) and the dispersion of EEC relations Equations (5) and (40), it is worth interpreting as anthropomorphic entropy properties [51] without penetrating its components.

Summarizing these considerations, it can be stated that:

The thermodynamic aspect of Equation (15) leads to EEC as shown in Figure 1, and at the same time the A&M equation is itself compensated by Equation (27).For this reason, the imperfection of Equation (31) for decalin and to a lesser extent for tetralin can be explained by deviations from thermodynamic values at melting temperature.It has been theoretically shown that approximating ∆*c_p_* with entropy is justified, but it is the average value of the population, understood as a set of only hydrocarbons, without a solvent.Linear correlation of the coefficients at the functional terms in Equation (15), *A*_1_ vs. $A_{2}$ determines the mean values of the enthalpy and entropy of the analyzed systems. Equation (40) determines the compensation temperature, except that the thermodynamic quantities refer to the mean values of the population, analogously to Equation (5), which determines melting temperature.

Thus, Equation (15) is richer in thermodynamic aspects, but further steps should be guided by the modification of the replacement of the activity by the mole fraction (*x*) and the determination of the activity coefficients using measured quantities (in terms of *T*, *x*).

### 3.4. Forms of Activity Coefficients in This Paper

The analysis of the subject shows the activity coefficients play an essential role in the analysis of the solubility processes. They were introduced to Equations (30), (31) and (33) and it makes sense to analyze whether they can be replaced in a different way, for example, by basic measured quantities, for the considered solutions in relation *x* vs. *T*.

For real, regular solutions with thermodynamic characteristics: $S^{E}=0,\;\;V^{E}=0,\;\;V^{m}=0,\;\;G^{E}=H^{E\;}\;$the measures of deviations is the excess thermodynamic potential expressed by:(41)$$G^{E}=RT\;\sum^{\;}x_{i}\ln\gamma_{i},$$

The quantities in Equation (41) can be determined by experimental or computational approximation, but the expression $(\gamma_{2}$ =γ) should be extracted from this equation and used in Equation (11), for example.

For sparingly soluble systems, it is necessary to know the Hildebrand solubility parameters *δ* [36,37,38]. 

For binary solutions ($x_{1}+x_{2}\equiv 1)\;$ excess free Gibbs energy is expressed by the formula of Schatchard & Hildebrand (S-H) [42,45,52,53,54,55]:(42)$$G^{E}=x_{1}x_{2}\left(\frac{V_{1}^{o}V_{2}^{o}}{x_{1}V_{1}^{o}+x_{2}V_{2}^{o}}\right){\left(\delta_{1}-\delta_{2}\right)}^{2},$$

or in extended version S-H, $l_{12}\neq 0\;$[42]:(43)$$G^{E}=x_{1}x_{2}\left(\frac{V_{1}^{o}V_{2}^{o}}{x_{1}V_{1}^{o}+x_{2}V_{2}^{o}}\right)[\left(\delta_{1}-\delta_{2}{)}^{2}+2l_{12}\delta_{1}\delta_{2}\right],$$

By expressing the number of moles in the place of Equation (41) instead of the mole fraction, we get [16,23]:(44)$$nG^{E}=RT\underset{i}{\sum }n_{i}\ln\left(\gamma_{i}\right),\;x_{i}=\frac{n_{i}}{n},\;T,\;P=\mathrm{const}.$$

making it easier to express molar excess Gibbs *i*-th component:(45)$$RTln\left(\gamma_{i}\right)=G^{E}+n{\left(\frac{\partial G^{E}}{\partial n_{i}}\right)}_{T,P,n_{j}},\;n_{i}\neq \;n_{j}$$

After returning to the molar fraction scale again and for the dissolved solid phase, i.e., for *i* = 2, we get:(46)$$RTln\left(\gamma_{2}\right)=G^{E}+\frac{\partial G^{E}}{\partial x_{2}}-\left(x_{1}\frac{\partial G^{E}}{\partial x_{1}}+x_{2}\frac{\partial G^{E}}{\partial x_{2}}\right),\;T=const$$

which can be recast as:(47)$$RTln\left(\gamma_{2}\right)=G^{E}+x_{1}\left(\frac{\partial G^{E}}{\partial x_{2}}-\frac{\partial G^{E}}{\partial x_{1}}\right),$$

Using Equation (43) and determining partial derivatives (*T* = *const*) we get:(48)$$RTln\left(\gamma_{2}\right)=V_{2}^{o}\Phi_{1}^{2}[\left(\delta_{1}-\delta_{2}{)}^{2}+2l_{12}\delta_{1}\delta_{2}\right],\;\mathrm{where}\;\Phi_{1}=x_{1}\frac{V_{1}}{x_{1}V_{1}+x_{2}V_{2}}$$

Assuming the simplest variant of a regular and simple solution, i.e., in Equations (42) and (43) we assume $V_{1}^{o}=V_{2}^{o}=V^{o}:$(49)$$G^{E}=B*x_{1}x_{2},\;\mathrm{where}\;\;\;B=V^{o}[\left(\delta_{1}-\delta_{2}{)}^{2}+2l_{12}\delta_{1}\delta_{2}\right]$$

After using Equation (49) in Equation (47) and after determining the partial derivatives, we obtain the sought relationship for $x_{2}=x,$ $\gamma_{2}=\gamma$:(50)$$\ln\left(\gamma \right)=\frac{B}{R}\frac{{\left(1-x\right)}^{2}}{T},$$

Relationship: ln(γ) vs. $\frac{{\left(1-x\right)}^{2}}{T}\;$for the version without intercept, is a directly proportional straight line with a slope (B/R)..

Table A2 presents the analysis of Equation (50) for the 12 hydrocarbons analyzed. In all the 24 cases $\rho^{2}>$0.999, except for naphthalene in tetralin ($\rho^{2}>$0.998), and this means the activity coefficients do meet the criteria for regular and simple solutions.

From the comparison of the constant B (according to Equation [49]) for tetralin and decalin it clearly indicates the excess potential is clearly higher for decalin, and carbazole is a clear deviation in both solvents.

In the light of the analyzed material, by combining Equation (15) for $A_{o}=0$ with an additional term of Equation (51), we obtain an extensive form of the A&M equation. The procedure reflected in the literature regarding adding additional members [46,47] was described in detail in the case of carbazole in 32 non-electrolytes and in water [46].
(51)$$\ln\left(x\right)=A_{1}\left(\frac{1}{T}-\frac{1}{T_{m}}\right)+A_{2}\ln\left(\frac{T}{T_{m}}\right)-C\frac{{\left(1-x\right)}^{2}}{T},\;\mathrm{where}\;C\neq \;\frac{B}{R},\;$$

Correlations of Equation (51) are shown in Table A3, in which the enthalpy is also given at melting temperature. The data are usable as previously, from formula Equation (20).

In several cases, unreliable values are obtainable. We find this results from too much dilution of the solution, which means that this effect occurs when: $\;x=x_{2}\ll 0.5.$

This is the case with anthracene and also acridine in tetralin and anthracene, acridine, thioxanthene and dibenzothiophene in decalin.

Because for regular solutions, $G^{E}=H^{E}\;$, because $S^{E}=0$, therefore Equation (43) can be represented in the form:(52)$$H^{E}=x_{1}x_{2}\left(\frac{V_{1}^{o}V_{2}^{o}}{x_{1}V_{1}^{o}+x_{2}V_{2}^{o}}\right)[\left(\delta_{1}-\delta_{2}{)}^{2}+2l_{12}\delta_{1}\delta_{2}\right],\;S^{E}=0,$$

and determine the excess enthalpy relationship vs. $x=x_{2}$.

By determining the constant *B* from Equation (49) with Equation (52) and inserting into Equation (50) we get:(53)$$H^{E}=\frac{x}{1-x}RTln\left(\gamma \right),$$

from which it follows that the excess enthalpy depends on the composition of the saturated solution and the known activity coefficient. For small values of *x*, the relationship Equation (53) is linear with respect to *x*, because $H^{E}≅\;xRTln\left(\gamma \right)$, and for higher values it is curvilinear.

The example is illustrated in Figure 5, which shows the solubility of anthracene in tetralin for which Equation (53) does not give correct results and carbazole in decalin, where the expected result compliance was observed.

The maximum value is per mole fraction $x_{1}=x_{2}$ = $\frac{1}{2}$ ($x_{1}*x_{2}$ = $\frac{1}{4})$ according to the data in Table A3 and Equation (49) for regular solutions $G^{E}=H^{E}\;$maximum enthalpy changes are from $H^{E}=$ 0 to $H^{E}$ = 921.76**R*/4 = 1.92 kJ*mol^−1^ (carbazole in decalin) so they are endothermic and small in relation to ${∆}_{m}H\;$(${∆}_{m}H\;\gg \;H^{E}$).

Attention was paid to the excess enthalpy of solutions. This size is not too large in relation to ${∆}_{m}H$ but it is variable and does not affect the essence of equations (coefficient of determination, level of significance), but on the enthalpy values determined at melting temperature.

For this reason, $H^{E}\;$ can be omitted, but their basic nature is due to variability to a varying degree depending on the value of the molar fraction of the substance dissolved at a given temperature to the state of saturation. If we use Equation (51), it is necessary to determine temperature curves with the maximum molar content of the dissolved component $x=x_{2}\gg 0.5$.

## 4. Discussion

Pan et al. [56] according to Krug [57] …“The compensation temperature is the temperature at which enthalpy variations precisely cancel entropy variations such that the rate or equilibrium constants are completely invariant.” In the case of research on solubility, a problem arises inspired by Freed [58], in which form a van ’t Hoff plot can be employed using the derived thermodynamic forms.

The statement about the equilibrium constant’s relative to the temperature of compensation requires commentary.

According to Equation (26), the role of such temperature for individual chemical compounds is fulfilled by $T_{m}$ and according to Equation (25), ∆G≡0. Therefore, it remains to consider the relationship of this temperature with the compensation temperature $T_{c}.$

Let’s assume that in Equation (3) we determine Gibbs free energy at the temperature $T_{c}$ and substitute the EEC linear relationship defined by Starikov [51,59]:(54)$$∆H=T_{c}\;∆S+Const,$$

We obtain
(55)$$∆G\left(T_{c}\right)=\left(T_{c}\;∆S+Const\right)-T_{c}\;∆S,$$

and in the absence of an intercept in Equations (54) and (55)
(56)$$∆G\left(T_{c}\right)=Const\;or\;0,$$

which means that the intercept in Equation (54) is an enthalpy that can be exchanged because it has an energy dimension, i.e., it is Gibbs free energy at *T_c_*.

The common point allows us to write:(57)$$∆G\left(T_{c}\right)={∆}_{m}H_{1}-T_{c}{∆}_{m}S_{1}={∆}_{m}H_{2}-T_{c}{∆}_{m}S_{2}=\cdots ,$$

and after summing and using Equation (40) we get the result according to Equation (56):(58)$$∆G\left(T_{c}\right)={∆}_{m}\bar{H}-T_{c}{∆}_{m}\bar{S},$$

Thus, it has been shown that EEC appears as an approach to the problem directly when we use the A&M equation for an individual chemical compound-solvent system, while in the general population common point is at melting temperature.

Grant et al. state that the non-linearity of van ’t Hoff’s isobar (ln(x)∝ 1/T) requires the use of three-parameter equations [60] that are also consistent with the chemical processes of thermal dissociation of the solid phase [61].

The question remains, however, to what extent model Equations (6) or (8) enable the determination of entropy. This is important because there are works proposing such a methodology.

Krug et al. propose the method described in [62], readily used for various systems in further works, including [12,30,40,50,63,64].

For the assumption *γ* = 1, *a = x* reasoning can be presented as a system of successive equations:(59)$${\left[\frac{\partial \ln\left(x\right)}{\partial \left(\frac{1}{T}-\frac{1}{T_{h}}\right)}\right]}_{P}=-\frac{∆H}{R},$$
(60)$$∆G=-RT_{h}*intercept$$
(61)$$∆S=\frac{∆H-∆G}{T_{h}}$$
where $T_{h}$ is the mean harmonic temperature, the intercept in Equation (60) is shaped in the system of functional scales ln(*x*) vs. ($\frac{1}{T}-\frac{1}{T_{h}})$, and Equation (61) is a special notation of Equation (3). Formally, thermodynamic functions relate to the temperature indicated by the index, but are constant in the considered temperature range.

By writing down Equation (59) for the experimental data and the "intercept" components resulting from the least squares method, we get:(62)$$\ln\left(x\right)=-\frac{∆H}{R}\left(\frac{1}{T}-\frac{1}{T_{h}}\right)+[\;\frac{\sum^{\;}\ln\left(x\right)}{N}-\left(-\frac{∆H}{R}\right)*\;\frac{\sum^{\;}\left(\frac{1}{T}-\frac{1}{T_{h}}\right)}{N}],$$

The last term in square brackets in Equation (62) is close to 0 (10^−7^–10^−8^), so the final form is as follows:(63)$$\ln\left(x\right)=-\frac{∆H}{R}\left(\frac{1}{T}-\frac{1}{T_{h}}\right)+\frac{ln\prod^{\;}x}{N}$$

Thus, Equation (61) is in a form with a mathematical structure typical of entropy:(64)$$∆S=\frac{∆H}{T_{h}}+R\bar{\ln\left(x\right)},\;\mathrm{where}\;\bar{\ln\left(x\right)}=\frac{ln\prod^{\;}x}{N}$$

It can be shown the identical form to that in Equation (64) can be obtained when we assume Equations (6) or (8), i.e., as: intercept = $\frac{∆S}{R}.$ This directly means that the intercept in these equations contains the desired entropy. For ease of use, in Equation (6) on the right we will extract the entropy in front of the expression and use Equation (5) to get Equation (13) for the activity version.

Using both Equations (6), (8) and (64) for both cases, the determined data are shown in Figure 6.

Figure 6 shows the formation of EEC for the population, with carbazoles and anthracene being distinct in both solvents. The evidence supporting such an effect was adopted as the criterion for determining the melting temperature according to Equation (5). The obtained data were grouped according to solvents—for tetralin, only cases deviating from the difference between temperatures ±1 K were shown, while for decalin, only positive ones according to the same findings (Table A4).

The A&M equation was used instead of Equation (11) for the considerations cited here to expose the dimensionless simplex relations ($T/T_{m})$ as an argument to the written-out function Equation (1). In particular, the coordinate [$\frac{1}{T_{m}};0]\;$is of great importance.

The A&M equation as Equations (15) and (31), together with Equations (20) and (21) as autonomously correlating thermodynamic functions through EEC, have an advantage over the approach Equation (6), but it is extremely important to respond to the need-to-know activity.

It is known that the simplest method of experimental determination of activity coefficients is the assumption of specific values of thermodynamic functions, such as ${∆}_{m}H$, or ${∆}_{m}S$ and $∆c_{P}$ and the use of Equation (11). Development of methods for determining these coefficients using UNIQUAC, UNIFAC or ASOG methods where only solubility curves ln(x) vs. temperature and the necessary physicochemical constants enrich the knowledge of thermodynamic properties of the tested systems. However, the question arises whether the reverse direction will allow us to determine selected thermodynamic functions. While the problem of determining basic thermodynamic functions is well-recognized, ignoring the coefficients of activity, it is interesting to connect the basics, captured by various forms of equations, with the part enabling the use of activity coefficients. Unprecedented forms of equations have been proposed due to the lack of knowledge of these coefficients, e.g., Equations (30), (31) or (33). As the example (Figure 5) shows, the analyzed regular solutions are not simple because $V_{1}^{o}\neq \;V_{2}^{o\;},$ but this fact does not affect the quality of the correlation.

To obtain the correct enthalpy values ${∆}_{m}H$ for analysis, use data on the largest possible share of dissolved solid phase, at least above the molar fraction > 0.5.

Equation (53) is restricted by right boundary conditions Equation (4) when *T$\;\to T_{m}.$*

This is due to the fact that for solutions that are too diluted, the activity coefficients are the highest, exceeding the frequently adopted value *y* = 1.

One more detail should be noted. In the analysis of the physical dissolution process, the priority importance of enthalpy over entropy is emphasized, which is expressed here by Equation (5). In contrast, Starikov et al. define the acronym EEC as Entropy-Enthalpy-Compensation [59,65,66,67], ascribing entropy to the source of this effect.

## 5. Conclusions

On the example of the thermodynamic analysis of the Apelblat & Manzurola equation, included in the Equations (15) and (28)–(33), it has been shown that expressing the absolute temperature through bifunctional ($\frac{1}{T},\;\ln T$) members is an alternative to the simultaneous determination of enthalpy and entropy. The ratio of this determines the melting temperature, in accordance with Equation (5), as $T_{m}={∆}_{m}H/{∆}_{m}S=∆H/∆S=const,\;$where the thermodynamic functions are temperature dependent (Figure 1). Thus, it has been shown that the structure of the A&M equation is itself thermodynamically compensated by EEC. Combined use of bifunctional functional members with respect to temperature is thermodynamically justified and brings new information in this regard.It has been shown that the simplest Equations (6) or (8) also allow for the simultaneous determination of enthalpy and entropy. Details of the transformation are included in Equations (59)–(64) but these quantities do not always come down to melting temperature in isoequilibrium state according to ($T_{m}).$It has been shown that approximation $∆c_{P}$ by entropy is justified, but it is an average value from the population, understood as a set of hydrocarbons (without solvent). Linear correlation of coefficients with functional terms in Equation (15), $A_{1}$ vs. $A_{2}$ determines the average values of enthalpy and entropy of the analyzed systems (Figure 3c).Saturated solutions of 12 polynuclear aromatic hydrocarbons, including those containing heteroatoms (N, S, O) in tetralin and decalin, have been described with A&M Equation (15). The end of the dissolution process is recognized by melting point temperature, i.e., for pure solute (without solvent) $x=x_{2}^{sat}\equiv$ 1. Equations (28)–(31) contain three or two factors and the dependent variable is in the form of concentrations or activities. A very good linear correlation was found for the equation coefficients Equation (15) at the functional members, expressed by Equation (37). In this way, Equation (40) defines the thermodynamic aspect of the compensation temperature—analogically acc. to Equation (5). It is the ratio of average enthalpy to entropy values in the analyzed population, for the considered population of 12 hydrocarbons in two solvents. As shown in Table A1, Table A2 and Table A3 and these calculations, the total numerical variability of molar enthalpy is 16.5–29.4 kJ*mol^−1^ (in [21]: 16.8–28.6 kJ*mol^−1^). The discussion presented for these equations indicates high compatibility of molar enthalpy with literature data [14,17,23].It has been proposed to extend the Apelblat & Manzurola equation in the form of Equation (51) after eliminating intercept and inserting in its place a characteristic segment for the coefficient of activity for regular and simple solutions Equation (50). Since in several cases unbelievable values were obtained, it was found that this is the result of too much dilution of the solution (Figure 5), which practically means that this effect occurs when: $\;x=x_{2}^{sat}\ll 0.5.$ It should be noted that there are no ideal conditions for this premise ($V_{1}^{o}\neq \;V_{2}^{o\;})$ but this fact does not affect the quality of the correlation.On the basis of simplified forms for regular and simple solutions, a significant problem is the variability in binary solutions of excess Gibbs free energy depending on the molar fraction of the solute. This applies to the acceptance of the adoption of such solutions, which are correct in the notation Equation (49), i.e., considerations based on the simplest approach in the Hildebrand theory.

### Computational Techniques

Calculations were carried out in the MS Excel using the REGLINP function taking accordingly determinations coefficient: $r^{2}$—two-parameter linear correlation, $R^{2}$—multiple correlation, $\rho^{2}$—correlation (linear, multiple) without intercept. At the same time, in the correlation calculations, the coordinate was added [$\frac{1}{T_{m}}$; ln(a)=0] or $[\frac{1}{T_{m}};\ln\left(x\right)=0].$

## Figures and Tables

**Figure 1 entropy-25-00055-f001:**
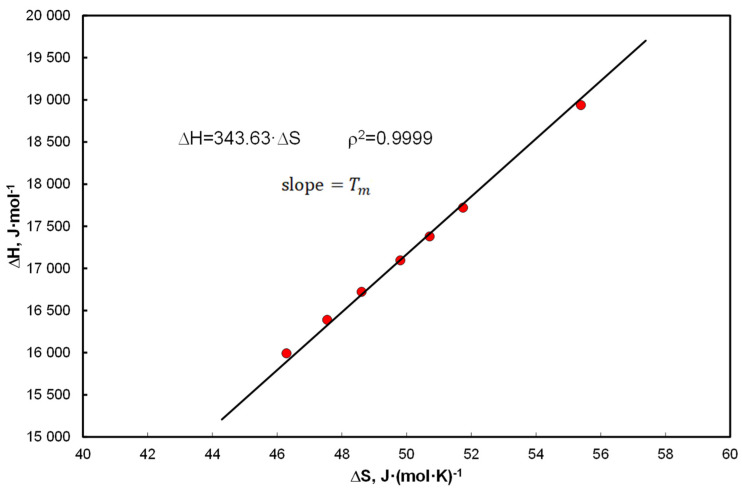
The EEC relationship for biphenyl in a saturated solution in tetralin for the coefficients Equation (31): $A_{1}$ = 1060.6 K, $A_{2}$ = 9.764 using Equations (19) and (24) and Equations (20) and (21) $T_{m}=\frac{∆H}{∆S}=\frac{{∆}_{m}H}{{∆}_{m}S}$.

**Figure 2 entropy-25-00055-f002:**
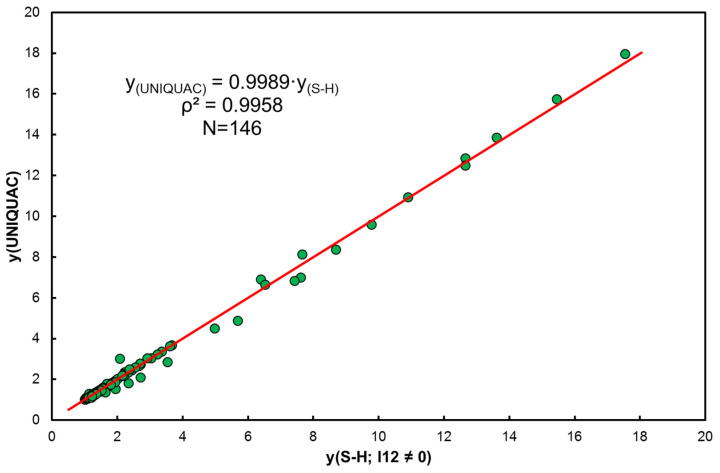
Comparison of activity coefficients according to UNIQUAC and extended S-H ($l_{12}\neq 0)$.

**Figure 3 entropy-25-00055-f003:**
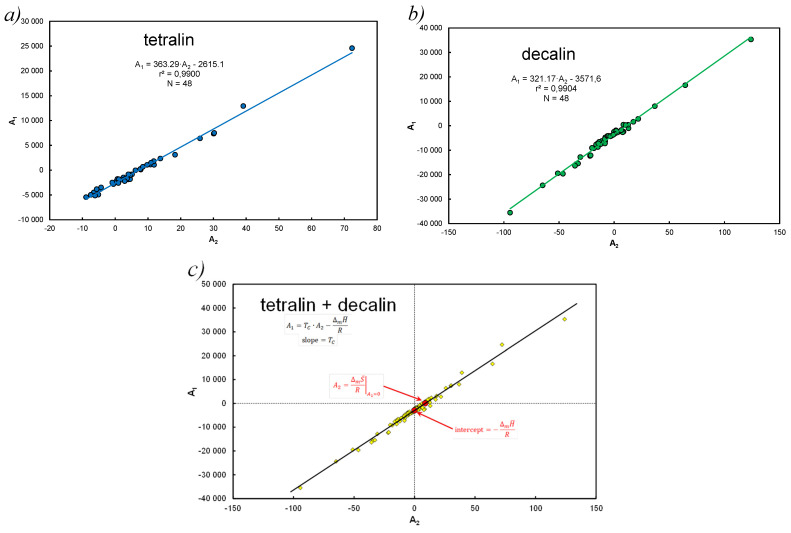
Correlation of A&M equation constants for considered solvents: (**a**) separately for tetralin, (**b**) separately for decalin, (**c**) together for tetralin and decalin (*A*_1_ = 334.09*A*_2_ − 2953.6, *r*^2^ = 0.985, *N* = 96).

**Figure 4 entropy-25-00055-f004:**
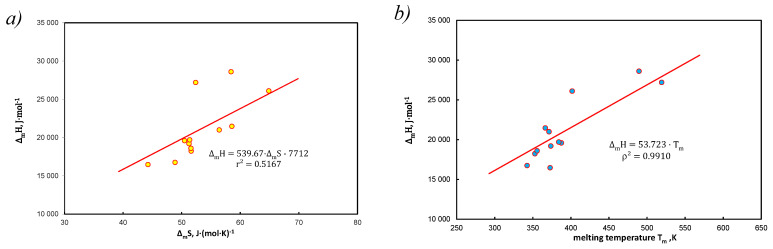
EEC diagram for 12 polynuclear aromatics at melting temperature, data from [21] (**a**) typical relation ${∆}_{m}$*H* vs. ${∆}_{m}S$, (**b**) different approach, slop determines average entropy, ${∆}_{m}\bar{S}$ = 53.7 J*(mol*K)^−1^, in accordance with [40].

**Figure 5 entropy-25-00055-f005:**
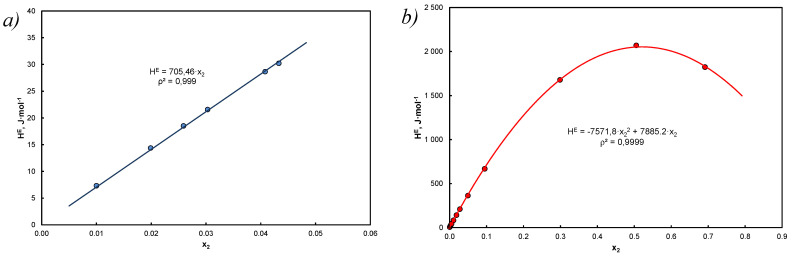
Comparison of excess enthalpy $H^{E}$ relative to the mole fraction according to Equations (52) and (53): (**a**) anthracene in tetralin, high dilution,$\;x=x_{2}$ ≤ 0.0433, for Equation (52): $V_{1}^{o}=$ 136.9 cm^3^ mol^−1^, $\delta_{1}$ = 19.514 $\sqrt{{\mathrm{Jcm}}^{-3}}$, $V_{2}^{o}=158.1\;$ cm^3^/mol, $\delta_{2}$ = 19.871 $\sqrt{{\mathrm{Jcm}}^{-3}}$, $l_{12}$ = 5.871*10^−3^; (**b**) carbazole in decalin, for the Equation (52) $V_{1}^{o}=$ 155.5 cm^3^ mol^−1^, $\delta_{1}$ = 17.711 $\sqrt{{\mathrm{Jcm}}^{-3}}$, $V_{2}^{o}=143.0\;$ cm^3^ mol^−1^, $\delta_{2}$ = 23.452 $\sqrt{{\mathrm{Jcm}}^{-3}}$, $l_{12}$ = 2.45*10^−2^, red lines concern Equation (53), points were calculated on the basis of Equation (52), the equations given in the graphs are approximations with respect to x_2_.

**Figure 6 entropy-25-00055-f006:**
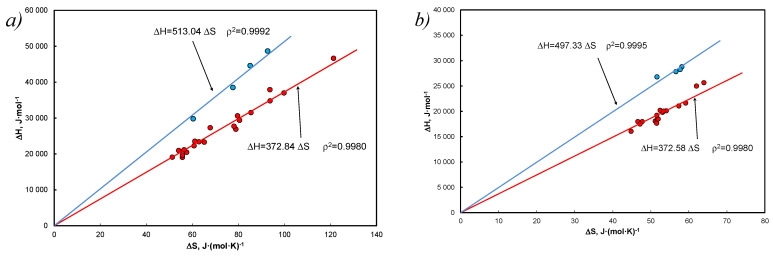
EEC according to the van ’t Hoff equation with the coordinate [$\frac{1}{T_{m}};0]$ and with the determined entropy for Equation (63) in form Equation (59), Equation (64) and (**a**) for Equation (8) and γ = 1, a = x, (**b**) for Equation (6) and activity (**a**). Blue colors refers to carbazole and anthracene in both solvents.

## Data Availability

Data taken from the literature [21]; in other cases, an appropriate reference was provided with the quoted data.

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
