# Peer review of "Enthalpy-Entropy Compensation Effect in Saturated Solutions on an Example of Polynuclear Aromatics According to Thermodynamics at Melting Temperature"

_entropy, 2022, doi:10.3390/e25010055_

Round 1

Reviewer 1 Report

GENERAL COMMENT: The manuscript reports the enthalpy-entropy compensation effect in saturated solutions on an example of 12 aromatic hydrocarbons in tetralin and decalin vs. temperature for saturated solutions. The Apelblat & Manzurola equation was used for extended model dissolution analysis. The manuscript is well written, but it´s important keep journal format.

1.       Abstract should be improved, avoid the use of equation numbers due to generated confusion to the reader. The abstract should be clear and provide the relevant information that summarize the research.

2.       The manuscript should be in format of the journal.

3.       A methods section should be included to clarify the information of the analysis done in the work.

4.       Figures and tables should be improved to the journal format. Guide lines in the figures can be deleted. Borders of the figures are not constant, please unify the format of the figures.

5.       Conclusion should be improved and adjusted to the relevant findings in the manuscript.

Reviewer 2 Report

Please find my comments below and in the attached file:

Dear editor,

In relation to the manuscript entitled “Enthalpy-entropy compensation effect in saturated solutions on an example of polynuclear aromatics according to thermodynamics at melting temperature” submitted by A. Mianowski and G. Labojko. The article assess the use of the Apelblat & Manzurola (A&M) equation for extended model dissolution analysis of 12 aromatic hydrocarbons in tetralin and decalin vs. temperature for saturated solutions  and its Enthalpy-Entropy thermodynamic compensation. The use of several equations, including the contributions of activity coefficients, is evaluated for the simultaneous determination of enthalpy and entropy. Generally, the paper is well written and the thermodynamics are postulated rigorously and introduced in a coherent progressive sequence in the very first part of the manuscript. However, as the manuscript introduces results, it becomes hard to follow perhaps for the excessive use of equations that are never used in the demonstration of the hypothesis postulated. I would suggest the authors to rewrite the manuscript in a much more concise way so that an easier to follow manuscript is submitted. I think that the manuscript fully falls within the scope of the journal and hence I recommend it publication provided that the mentioned change, and the specific comments detailed below are addressed.

Specific comments:

Keywords. Please reduce the number of keywords

Line 42. Please correct on the one hand to “on one hand”.

Line 88, please add subindex j to indicate that the activity, molar fraction and activity coefficients are related to one of the components j  in the mixture.

Line 213. What “czyli” means?

Why after page 13 the line numbers disappear?

Please provide a clear definition of the compensation temperature in thermodynamic sense in page 15, and the implications of its value.

Please amend the y-axis labelling in Figure 5.

The way in which Figure 6 is presented is absolutely inappropriate, e.g. too much blank space, no legend for identifying the different colours of the series, use of same symbols, etc. Please improve the artwork of this figure.

It is weird that the ln (a) in Fig. 6 do not approach asymptotically to zero as T tends to Tm.  The meaning of that is that the proposed equation does not show a coherent temperature dependence close to Tm, which somewhat questions the thermodynamic rigout of the modified A&M equation. Any explanation for that?

Please use different colours and symbols for Figure 7 to better distinguish the different series and their meaning.

How were the solubilities used in the calculations determined. This must be properly specified and referenced within the text.

Rewrite the conclusions avoiding the excessive mention to numerical equations within the text and better highlight the meaning of the conclusions extracted. Many of them are not even conclusions.

Pag 17. A big inconsistency is considering that the hear capacities of the solid and the supercooled melts are constant with temperature. I suggest the authors to perform proper DSC analysis to account for these variation that is often bigger than 100 J/molK in a temperature range of 100 C.

Reviewer 3 Report

The authors apply known models and modifications of these models to experimental thermodynamic literature data of aromatic hydrocarbons dissolved in tetralin and decalin.

The work performed here is of interest to the community but the manuscript needs to be reorganized and heavily edited prior to publication.

The inclusion of equation numbers referencing the document in the abstract (lines 18, 19, and 21) is confusing and makes the abstract difficult to read without referencing the manuscript.

The section "Equations describing saturated solutions in terms of temperature" appears to be known from the literature. It should be condensed, summarized, or placed in the supporting information.

The statement on line 111 requires further support and explanation.

Line 118 states "Omitting the signs in Eq (14)" referencing Eq 15 but the signs from Eq 14 persist in Eq 15. The meaning of this should be clarified.

The "aims" statement in line 163-166 is good.

Line 175 references "version ln(a)" for Eq (15) but there is no ln(a) term in Eq 15. This must be clarified.

Figures are inconsistently formatted and need to be standardized.

Sources for the data and a statement of exactly what data was used for each source should be added to the "Data Availability Statement".

Reviewer 4 Report

Comments on entropy-2036708:

The current work by Andrzej Mianowski and Grzegorz Łabojko reports an interesting investigation of the enthalpy-entropy compensation effect. Although the content could be suitable to the ‘entropy’ audience, the presentation and language require significant improvements.

Many abbreviations are defined but not mentioned later. For instance, EEC is defined in the abstract but is never cited later. In that case, they should not be defined at all. On the other hand, many abbreviations are defined multiple times. An example is still EEC, which in the main text is defined on page 2 for the first time and on page 6 for the second time. The authors should really carefully check this issue.

The abbreviation Eq and the full name equation are used in the same article, which seems informal in scientific publishing. Further, ‘Eq’ as the abbreviation of equation should be written as ‘Eq.’ in the main text.

For the correlation coefficient reported in plots, I identify r2 and rho_2 used in the same article. This should also be made consistent.

There should be some uncertainty estimates for physical observables, but these statistical errors are not really reported for many properties, e.g., Table 1.

I wonder whether the linear regression reported in this paper has considered the uncertainty of each point in fitting. Normally, data points with larger uncertainties should be assigned smaller weights in the loss function.

On page 3, free Gibbs energy -> Gibbs free energy

Honestly, the formatting issue makes reviewing rather painful. There are many problematic points not mentioned in the above detailed comments. I strongly recommend the authors to carefully proofread the manuscript and correct all existing formatting issues.

Round 2

Reviewer 2 Report

The authors have improved the quality of the article substantially and it could be accepted for publication in its current form

Reviewer 3 Report

The authors have adequately addressed my concerns and the paper should be ready for publication after minor revisions to the writing for brevity.

Reviewer 4 Report

Acceptable in the current form.